# Portable Device for Multipurpose Research on Dendritic Yanson Point Contacts and Quantum Sensing

**DOI:** 10.3390/nano13060996

**Published:** 2023-03-09

**Authors:** Andriy Savytskyi, Alexander Pospelov, Anna Herus, Volodymyr Vakula, Nataliya Kalashnyk, Eric Faulques, Gennadii Kamarchuk

**Affiliations:** 1B. Verkin Institute for Low Temperature Physics and Engineering, 47 Nauky Ave., 61103 Kharkiv, Ukraine; 2National Technical University “Kharkiv Polytechnic Institute”, 2 Kyrpychov Str., 61002 Kharkiv, Ukraine; 3Université de Lille, CNRS, Université Polytechnique Hauts-de-France, UMR 8520-IEMN, F-59000 Lille, France; 4Institut des Matériaux de Nantes Jean Rouxel, Nantes Université, CNRS, IMN, F-44000 Nantes, France

**Keywords:** quantum sensor device, conductance quantization, Yanson point contact, cyclic switchover effect, dendrite, Raman spectroscopy

## Abstract

Quantum structures are ideal objects by which to discover and study new sensor mechanisms and implement advanced approaches in sensor analysis to develop innovative sensor devices. Among them, one of the most interesting representatives is the Yanson point contact. It allows the implementation of a simple technological chain to activate the quantum mechanisms of selective detection in gaseous and liquid media. In this work, a portable device for multipurpose research on dendritic Yanson point contacts and quantum sensing was developed and manufactured. The device allows one to create dendritic Yanson point contacts and study their quantum properties, which are clearly manifested in the process of the electrochemical cyclic switchover effect. The device tests demonstrated that it was possible to gather data on the compositions and characteristics of the synthesized substances, and on the electrochemical processes that influence the production of dendritic Yanson point contacts, as well as on the electrophysical processes that provide information on the quantum nature of the electrical conductance of dendritic Yanson point contacts. The small size of the device makes it simple to integrate into a micro-Raman spectrometer setup. The developed device may be used as a prototype for designing a quantum sensor that will serve as the foundation for cutting-edge sensor technologies, as well as be applied to research into atomic-scale junctions, single-atom transistors, and any relative subjects.

## 1. Introduction

Quantum structures are ideal objects by which to discover and study new sensor mechanisms and implement advanced approaches in sensor analysis to develop innovative sensor devices [1,2,3]. They are able to provide the necessary conditions for the development of sensor technologies based on new principles that are fundamentally different from the methodology currently used to create traditional sensor devices. Among the wide variety of sensors, chemical sensors using the principle of electrical conductance changing under the action of an external agent attract the most attention [4,5,6]. These devices owe their popularity to their unique parameters, which allow the implementation of highly efficient sensor technologies. Among the advantages of chemical sensors, one can note the high speed of operation, which allows real-time measurements, the portability, which makes them easy to use and guarantees the ability to quickly move them where they are needed, the high energy efficiency, which ensures offline operation without involving external sources of energy, the low cost of production, and the high economic efficiency of operation just to mention a few. Sensor technology has become one of the most dynamic and promising areas in modern research and development. It has already contributed to the creation of a large variety of sensor devices, including nanosized ones, and their everyday application. Using modern nanosensors to detect different gases and liquids is now becoming a common practice [7,8]. Control over gaseous and liquid media is of great importance for both the reliable operation of many technological cycles and the protection of the environment [9,10,11].

Quantum structures and quantum sensor mechanisms make it possible to elevate the technologies for the development of sensor devices to a qualitatively new level [12]. In particular, this approach opens up unlimited possibilities for using the most common and readily available materials to create sensors. For example, traditional metals that were previously of little or no interest to sensor developers can now turn into high-tech objects and inexhaustible sources of raw materials for the next generation of quantum sensors. A typical example is an ordinary copper, which has proven itself in the creation of dendritic Yanson point contacts [1,12]. This allows the implementation of a simple technological chain to activate the quantum mechanism of selective detection in gaseous and liquid media. It includes a series of technologically easily implemented operations, such as choosing copper as a material for the bulk electrodes involved in the synthesis of dendritic Yanson point contacts [13], launching the electrochemical cyclic switchover effect in the system of copper electrodes [14], creating nanostructures in the form of dendritic Yanson point contacts, observing quantum effects in their electrical conduction, and using them for selective detection in gases and liquids [1]. The mechanism of selective detection in gases is based on the energy principle of object recognition. Each quantum system is characterized by a set of quantum states that have a certain energy. Knowing this set allows one to unambiguously identify the analyzed object. In the process of the electrochemical cyclic switchover effect, quantum transformations take place in the structure of the material of the conduction channel of the dendritic Yanson point contact, which makes it possible to observe and record quantum states in the electrical conductance of this unique nanostructured object. Moreover, it turns out that the involvement of a gas or liquid in this process leads to a characteristic change in the set of quantum states of the initial system [1,14]. As a result, the set of these states becomes the key to identifying the agent to be detected.

At the initial stage of work on the discovery and study of the electrochemical cyclic switchover effect [14], a modified device was used, a prototype of which usually serves to produce the needle–anvil point contacts in the Yanson point-contact spectroscopy [15]. During the work, it turned out that in order to understand the complex quantum nature of the electrochemical cyclic switchover effect, it is necessary to be able to simultaneously obtain information (at least):-about the electrochemical processes affecting the synthesis of dendritic Yanson point contacts;-about the electrophysical processes which can shed light on the quantum nature of the electrical conduction of dendritic Yanson point contacts; and-about all phases of the process, including control, visualization, and analysis of both the synthesis itself and the byproducts formed during the synthesis of dendritic Yanson point contacts.

To solve these problems, and to develop a methodology for promptly launching and monitoring the electrochemical cyclic switchover effect, thus providing the necessary conditions for its comprehensive study and for the development of elements of the technology for its subsequent exploitation, it is necessary to have a device that allows multipurpose research with the possibility of applying various in situ methods—that is, simultaneously recording signals of different nature in a multichannel mode. At this stage of study of the physicochemical processes occurring during the electrochemical cyclic switchover effect, the information that is necessary for analysis of the processes of transformation of materials at the atomic and mesoscopic levels, identification of sensor effects, and development of quantum sensors can be obtained by using electrophysical, electrochemical, and optical methods of research.

The discovery of the quantum mechanism of selective detection in gaseous and liquid media, as well as the wide opportunities for the development of innovative approaches to designing quantum sensors, the operation of which is based on the original principle of detection in dynamic mode that we proposed in Ref. [1], creates demand for development of new devices and technological elements of the future quantum sensor technologies.

This predetermined the aim of this work, which is to design a device that provides simultaneous measurements in a three-channel mode for recording electrophysical, electrochemical, and optical parameters of the quantum processes and objects under study. At the same time, taking into account the prospects for the subsequent use of the device as a prototype of sensor, the device should be small in size—that is, meet the requirements of portability.

To our knowledge, the proposed device is a prototype developed for the first time for the realization of a quantum sensor, the operation of which is related to the new mechanism based on conductance quantization. The prospective pilot sample can be used to realize a series of new original quantum and sensing properties of Yanson point contacts and new effects arising from them. The quantum nature of the point-contact sensor response based on conductance quantization realized in the proposed device will ensure a successful search for novel quantum mechanisms of sensor analysis that have no analogues in the field of sensor technique, as well as their subsequent study, development, and implementation. The simplicity of construction and low cost of the point-contact sensor device will also stimulate its wide application in sensor research into the detection of different gaseous agents. It will establish a solid background for the development of principally new nanomaterial-based technologies and devices.

## 2. Technological Requirements

In order for the developed device to provide comprehensive capabilities for performing a full set of operations in the above technological chain for the implementation of the quantum mechanism of selective detection in gases and liquid media [1], it is necessary to fulfill a number of technological conditions. First of all, the device must provide the ability to create Yanson point contacts at a level comparable to the technological level of the Yanson point-contact spectroscopy [15,16]. For this to be done, there must be a mechanism for producing Yanson point contacts. This mechanism should be characterized by low cost, ease of production, and reliability in operation. The prototypes of such a mechanism are well known to the experts in the field of Yanson point-contact spectroscopy [12,15,16], because they are an integral part of the technology. It should be noted that the composition of the mechanism elements and its structure are determined by the method used to produce Yanson point contacts. To observe and study the electrochemical cyclic switchover effect, it is most convenient to work with contacts created using the “needle–anvil” method and Chubov displacement technique [12,17]. The technology of the Yanson point-contact spectroscopy implies the use of specially designed devices to implement these methods [15,18]. Their operation is based on the principle of converting rotational motion into translational motion. This principle is used by the electrode movement control system. This approach allows us to simplify the design of the control unit located outside the device to ensure the transfer of the angular momentum to the electrodes in order to create contacts inside the device. The connection between the external controls and the electrodes inside the device is provided by a system of cylinders that rotate around their own axis independently of each other. This makes it possible to implement the transfer of the angular momentum through standard small apertures in the device body, using the highly efficient and, at the same time, simple technological elements of the Yanson point-contact spectroscopy [15].

The main element of the mechanism for implementing the needle–anvil method and Chubov displacement technique in the Yanson point-contact spectroscopy is the differential screw [18]. Its advantage is the possibility of moving the electrodes, between which a contact is created, over long distances from each other. Usually, depending on the conditions of the experiments, the magnitude of such displacements can reach 10–20 mm, which is a huge value, given that the mechanism is focused on creating structures in the nanometer range. At the same time, when used in devices in the Yanson point-contact spectroscopy a differential screw is combined with a gearbox, an extremely small displacement of the electrodes can be ensured with a step of 25 nm [19]. As a result, when using the needle–anvil method, the researcher can easily place the electrodes at a distance which is needed to grow a single dendrite and launch the electrochemical cyclic switchover effect. This technological feature is very important for the creation of dendritic Yanson point contacts. In order to ensure the maximum concentration of the electric field in the small area between the tip of the needle and the anvil, the electrodes must be located at a minimum distance from each other. This guarantees the preferential growth of a single dendrite (Figure 1) followed by the creation of a dendritic Yanson point contact at the point of contact between the dendrite tip and the anvil.

The next requirement for the device is the high airtightness of all its components and connections. Its fulfillment makes it possible to control the experimental conditions inside the research cell and fine tune them, which is very important for work in environments with low concentrations of the analyzed substance. The preparation of the experiment requires creating a vacuum inside the cell for the total degassing of the cell volume and using distilled (ultrapure) water as a medium in which dendritic Yanson point contacts are produced. The device for multipurpose research on dendritic Yanson point contacts and quantum sensing must have a certain number of inlets that provide the transfer of the angular momentum to the electrodes to create the contacts, electrical connection with the electrodes, visual control of the processes occurring in the device, and connection of the cell volume to the vacuum pump, which is capable of creating a vacuum of 10^−2^–10^−3^ mm Hg. To make the device airtight, it is convenient to use standard vacuum seals, which are widely used in the Yanson point-contact spectroscopy [15].

Because the device is meant to be focused on the study and registration of various chemical agents, including those with a high chemical activity, its elements must be made of chemically neutral materials. These materials must have a low adsorption capacity and be characterized by the absence of porosity in order to avoid the accumulation of analyzed substances and reaction products inside the test volume. Stainless steel, duralumin, silicone and polyurethane seals, and inert plastics such as fluoroplastic can be considered for these purposes. In addition, the body of the device must provide resistance to excessive loads and deformations, for example, during the creation of a vacuum inside the cell. Based on this, preference should be given to a metal case. We chose duralumin of the D16T brand as the material of the device body. To ensure chemical resistance, the manufactured duralumin elements should be anodized in order to increase the thickness of the oxide layer because aluminum oxide is quite strong and fairly inert.

Adaptation of the device for multipurpose research on dendritic Yanson point contacts to the possibility of direct observation and registration of the dendrite growth/dissolution and the results of its synthesis by using optical devices requires matching of the cell dimensions with the parameters of the existing devices. These adjustments concern, first of all, the geometric dimensions limited by the cell height. Solving this problem will simultaneously ensure the fulfilment of the requirement of portability for using the device as part of the sensor technologies. Taking this requirement into account, the form factor of the device was inspired by the environmental and mechanical control stages compatible with microscope-based spectroscopy systems [20,21]. To enable in situ Raman investigations, an optical window was incorporated into the lid of our stage. It is possible to cover this window with sapphire and quartz glasses. Al_2_O_3_ is widely used in protective optical windows, sight glasses, prisms, and optical lenses in a wavelength range of 190–4500 nm. Sapphire demonstrates a unique combination of a wide transmittance range (from ultraviolet to infrared) and high transmittance, as well as strength and durability. According to the results of the market analysis, it was decided to use sapphire glass.

## 3. Device and Elements of the Exploitation Methodology

The laboratory prototype of the device for multipurpose research on dendritic Yanson point contacts and quantum sensing is made in the form of a quasi-rectangular block with characteristic dimensions of 10 × 15 × 3 cm^3^. The general view of this device is shown in Figure 2. The device is equipped with needle and anvil manipulators, with which the electrodes can be moved toward each other and back, thereby forming a Yanson point contact. The viewing window allows observation of the process of creation of a contact, which greatly simplifies its control, and also makes it possible to study with a micro-Raman spectrometer the reaction products in a given environment, which can be formed thanks to the airtight gas connectors.

To increase the capabilities of the device when conducting a variety of studies and to ensure its quick adjustment to work in new conditions, the inside of the device is made in the form of interchangeable cartridges. The development of the cartridge prototype was carried out on the basis of a 3D model of the device synthesized by us. This approach allowed us to select the optimal design solutions for using various types of manipulators to create point contacts used in the Yanson point-contact spectroscopy. This applies primarily to the possibility of rapid implementation and interchangeability of designs by using the needle–anvil method [12] (Figure 3a) or Chubov displacement technique [17] (Figure 3b) to produce Yanson point contacts. In addition, the 3D model of the cartridge (Figure 3c) made it easy to calculate the dimensions of the system components, electrical and vacuum switching channels, and their relative position.

Figure 3c shows a model of a cartridge equipped with holders for creating Yanson point contacts by using the Chubov displacement technique. When rods 1 and 2 rotate, the electrodes in the form of metal prisms fixed in holders 3 and 4 can approach each other or move away. This allows us to form a point contact 7, which is located in the cuvette 5. The design peculiarity is that the axes of the manipulator rods are at an angle of 10 degrees to each other. By moving both holders along the cuvette, we get the opportunity to create contacts between different parts of the electrode faces. This makes it possible to create many high-quality Yanson point contacts under identical conditions in the course of one experiment. Capillary 6 is located at the bottom of the cuvette. It fills the cuvette with pre-prepared electrolyte or pure water and then removes it. After the electrolyte solution is removed, a thin film of the liquid under study remains on the surface of the electrodes, which makes it possible to launch the cyclic switchover effect. Moreover, the capillary can be used to supply gas and blow various gases and their mixtures through the electrolyte in the cuvette.

Preparation and operation of the device for multipurpose research on dendritic Yanson point contacts and quantum sensing when using the needle–anvil method can be presented as follows. Before the experiment, the electrodes for creating the contacts are mounted inside the device. Then the device is closed and pumped out with a vacuum pump. This operation is needed to clean the internal space of the device by removing components of the environment and potential substances remaining there after previous studies. The duration of the pumping process is 1–5 min. After that, the device cell is additionally cleaned with a flow of an inert gas, for example, argon, and a drop of distilled water prepared for the study of electrolyte solution or liquid is put into the place of the planned contact between the needle electrode and the anvil through the capillary of the gas connector. When registering sensor effects or studying the interaction of dendritic Yanson point contacts with a gaseous medium, the corresponding gas agent is introduced into the internal space of the device. Studies can be carried out both in a stationary gaseous medium and in the flow of the analyzed gas. Then the device is connected to a setup with measuring instruments, which allows measurements in a three-channel mode to record electrophysical, electrochemical, and optical parameters of the studied quantum processes and objects. After that, the needle electrode is moved toward the anvil electrode until a point contact with direct conduction is created. The process of creating a contact is controlled by the electrical resistance of the “needle-distilled water–anvil” system. A point contact is formed when the resistance of the system falls below the resistance of a single-atom Yanson point contact of 12.9 kΩ. Note that the electrical conductance of a single-atom contact corresponds to an elementary conduction channel with the conductance value of one quantum. The launch of the cyclic switchover effect can be started both at the moment of formation of direct electrical contact and after mechanically breaking the initially created contact and placing the needle at the minimum distance from the anvil.

Let us first consider the latter case. After mechanically breaking the initially created contact, one can move the needle electrode away from the anvil with a step of 25 nm. One step of the manipulator is equal to one turn of the rod at the gearbox and it provides a distance between the electrodes that is enough to launch the cyclic switchover effect. We found that the growth of a single dendrite can preferably take place at distances up to hundreds of nanometers between the needle and anvil electrodes (Figure 1). Within this range, it does not matter what distance is chosen to launch the cyclic switchover effect. It should be noted, that the electrodes, especially the needle, are thoroughly treated and prepared for experiments—that is their side surfaces are clean and smooth and do not contain impurities or inhomogeneities that can provoke dendrite growth.

In the former case, one can create a Yanson point contact by using the methodology described elsewhere (see, for example, Refs. [12,15,16,18,19]). After the creation of the point contact, it is necessary to check its mechanical stability, and then, if it is sufficient, it is possible to launch the cyclic switchover effect by sweeping current through the contact. This technique is thoroughly described in Ref. [22]. Once the process of the cyclic switchover effect is started, it becomes possible to register the characteristics of the quantum processes of synthesis of dendritic Yanson point contacts. Depending on the tasks, the research can be either focused on studying the quantum nature of the effects or aimed at selective detection in gaseous or liquid media, which is based on the quantum mechanism discovered in Ref. [1].

## 4. Device Tests, Results, and Discussion

The device for multipurpose research on dendritic Yanson point contacts and quantum sensing was tested in atmospheric air. The aim of the first test concerning device functionality is the visualization of the processes under investigation and demonstration of the possibility to obtain direct information about the synthesis products and transformation of the materials in the working area. The research methodology corresponded to that described above. The distance *d*_e_ between the tip of the needle electrode and the anvil electrode during the electric current flow in the two-electrode system was shown to have a substantial impact on the nature of the electrochemical processes.

This discovery was made possible by the subsequent in situ optical investigations and the ability to directly observe the area of contact creation through the viewing window in the device body.

We found that if the needle (cathode) and anvil (anode) are sufficiently apart at the start of the experiment, we have conditions that are identical to those in a typical two-electrode electrochemical cell [23,24,25,26]. This distance is roughly a few microns on the scale typical of Yanson point contacts. In this instance, dendrites develop over the course of a lengthy experiment on the side of the needle that is not involved in the establishment of dendritic Yanson point connections or direct interelectrode conduction. The lateral side of the electrode develops a “coat” (Figure 4). The direction of the dendrite growth coincides with the electric field lines.

We anticipated that nanocrystals of copper and copper oxide are possibly present at the apex and lateral surface of the tip because of the formation of nanodendrites resulting from the electrochemical reactions during the dendrite synthesis. Raman spectroscopy can detect these compounds. We have used our stage as a demonstrator for identifying the substances created during the electrochemical experiments (Figure 5). Examples of the Raman results acquired with the Renishaw setup are displayed in Figure 6.

Micro-Raman experiments were performed in the device with a Renishaw inVia Raman microscope spectrometer (Figure 5) including a detector working with a thermoelectric chip cooled at −70 °C. Measurements were made at room temperature in the wavenumber range between 500 and 4000 cm^−1^. A laser excitation of *λ*_exc_ = 633 nm (1.96 eV) was provided by a helium-neon laser. The detector allowed registration of the low-frequency part of the spectra (10–150 cm^−1^) and some anti-Stokes Raman lines by using a Bragg reflector for this wavelength. The diameter of the laser spot on the sample surface was about 2 µm for the fully focused laser beam with 50× objective magnification. Before the experiments on the dendrite synthesis, the tip of the anvil electrode was generally found to be homogeneous under a microscope without an extraneous microcrystalline phase. The Raman instrument was calibrated against the Stokes Raman signal of pure Si at 520.5 cm^−1^ by using a silicon wafer. The spectral resolution was 2 cm^−1^. In each optical experiment, several spectra were recorded on different spots in the samples.

Table 1 lists the Raman frequencies obtained from the material covering the tip apex, those described in the literature for copper oxides, and those estimated with density functional theory (DFT) [27,28,29,30]. For this calculation, we employed the generalized gradient approximation (GGA) with the Perdew–Burke–Ernzerhof (PBE) functional allowing efficient computational cost and flexibility for calculating various physical properties. Ultrasoft pseudopotentials were used for the two crystals. Geometry optimization was carried out using the Broyden–Fletcher–Goldfarb–Shanno (BFGS) and density mixing optimization schemes with a force convergence threshold of 10^−4^. We implemented the relaxation of both cell parameters and atomic positions. The cutoff energy was fixed at 440 Ry (1 Ry = 13.60568 eV) for Cu_2_O and 600 Ry for CuO. Most of the experimental Raman modes observed up to 620 cm^−1^ on the materials covering our point-contact tips correspond well to CuO and Cu_2_O oxides signatures supported by experiments in literature and DFT calculations. We conclude that the dendritic material in our experiments can be identified as a mixture of these two oxides. However, we cannot yet ascribe several bands at 51, 473 cm^−1^, between 1000 and 1600 cm^−1^, and approximately 2900 cm^−1^ that we found in the tip spectra to another copper phase. We also note that Debbichi et al. [31] have recorded the Raman spectrum of Cu_4_O_3_, and found lines at 318 (m), 510 (w), 541 (s), and 651 (w) cm^−1^. Actually, no such lines were noticed in our spectra. Additionally, we have observed several infrared bands of copper oxides in the Raman spectra of the tips that were probably activated by structural defects [32,33,34,35,36].

The next portion of data that can be obtained by using the proposed device is information about the nature of the homogeneous and heterogeneous physical–chemical processes taking place during the cyclic switchover effect. Because the reactions that occur in the system have characteristic times at the level of milli- and microseconds, it is most useful to examine the characteristics of electrochemical processes in the system under investigation by using relaxation methods. It is well known that the most informative relaxation technique is impedance spectroscopy [37]. With the device under consideration, it is feasible to fully utilize this technique and collect a set of data for the investigation of the processes that are taking place.

Therefore, in addition to the Raman spectrum, an impedance spectrum was measured for the needle–electrolyte–anvil system in the frequency range of 100–200,000 Hz. The measurements were performed by using a VERSASTAT4-500 potentiostat (AMETEK, USA). Figure 7a shows the experimental (blue curve) and theoretical (red curve) frequency dependences of the active and reactive components of the system impedance. The reactive components of the system impedance for each of the frequencies are plotted on the ordinate, while the active components of the impedance for the same frequencies are plotted on the abscissa. Thus, in the presented coordinates, we experimentally obtain a frequency profile of this system. The essence of the electrochemical impedance method is that the system under study is modeled by a certain set of electrical elements and each element is assigned a certain characteristic or parameter (Figure 7b) [37]. This makes it possible to predict and track the subsequent behaviour of the system when its parameters and experimental conditions change. For example, active resistance included in the circuit in series with other elements reflects the resistance of the electrolyte located between the electrodes. Capacitance and resistance connected in parallel characterize the capacitance of the electrical double layer on the electrode surface and the resistance of the Faraday processes, respectively. To check the correctness of the simulation, an analytical expression is found for the impedance of the electrical circuit and explicit expressions are obtained for the frequency dependences of the active and reactive components for certain parameters of the elements. If the theoretical and experimental curves in the specified coordinates agree well, it is said that an adequate model of the system under study has been synthesized.

As expected, the hodograph obtained in our experiment has a form which is characteristic of systems in which electrochemical processes play a significant role. The diameter of the semicircle in the high-frequency region (Figure 7a) corresponds to the resistance of the electrochemical reactions, and extrapolation to the abscissa axis at high frequencies (that is, the projection of a point on the curve onto the abscissa axis) makes it possible to determine the electrolyte resistance. In the system under study, the Faraday processes can occur both at the tip of the needle and at the counterelectrode (“anvil”). Because the auxiliary electrode is much larger than the working one, almost all the measured resistance is determined by the processes occurring at the tip of the needle. Processing of experimental data using the Ershler–Randles method [38,39] allowed us to estimate the parameters of the equivalent circuit. It turned out that the resistance of the electrolyte is *R*_E_ = 4000 Ohm, the capacitance of the double electric layer at the border.

“needle point-electrolyte” is *C*_d_ = 5∙10^−10^ F, the Warburg constant is *W*_F_ = 10^6^ Ohm s^−0.5^, and the resistance of the Faraday processes is *R*_F_ = 30,000 Ohm. By using the obtained capacitance value, it is possible to estimate the active surface of the electrode, which participates in the electrochemical reactions, if we assume that the specific capacitance of the double layer is about 20 μF/cm^2^ [40]. Moreover, knowing the resistance of the electrochemical processes, it is possible to estimate the exchange current density at the metal–electrolyte interface. It turned out to be equal to 180 A/m^2^. The exchange current density is a fundamental characteristic of the electrochemical activity of electrodes. Knowing this parameter, we can synthesize analytical models of the electrode processes and predict the behaviour of complex electrochemical systems that underlie the operation of the quantum sensors we propose. The specific values of the obtained parameters are in good agreement with the literature data [41], which indicates the correctness of the simulation and the reliability of the presented results.

The purpose of the third test is to demonstrate the device operation in the regime of cyclic switchover effect and obtain data about conductance quantization that can be used for detection in liquid and gaseous media. The situation is different when the distance between the needle (cathode) and the anvil (anode) is small enough. On the scale typical of Yanson point contacts, this distance is approximately several dozens or hundreds of nanometers. Under this condition, a dendrite starts to grow on the tip of the needle electrode (see Figure 1), creating a Yanson point contact when it reaches the anvil electrode, and the cyclic switchover effect is observed [14]. The cyclic switchover effect consists of consecutive cycles of growth and dissolution of the dendritic nanochannel of the Yanson point contact as electric current flows through the “needle electrode–electrolyte–anvil electrode” system. It should be noted that the cyclic changes are automatic with no external influence. The detailed information about the quantum transformations in the crystalline structure and the electric conduction in dendritic Yanson point contacts during the cyclic switchover effect can be obtained from the dependence of resistance *R* on time *t*. One of the curves *R*(*t*) measured during the device tests is presented in Figure 8a.

During the cyclic switchover effect, cyclic changes in the crystalline structure and quantum electrical conductance of dendritic Yanson point contacts occur. During one cycle, the electrical resistance of the Yanson point-contact changes, which manifests itself in the *R*(*t*) curve in the form of areas of growth, decrease, and stabilization. Stabilization of the electrical resistance of the dendritic Yanson point contact reflects the emergence of a metastable quantum state with a certain energy. Metastable quantum states appear as sections of constant conductance looking like steps in the dependence *R*(*t*) (Figure 8b). They reflect the quantization of conductance and the electronic shell effect, which is responsible for the formation of the crystalline structure during the growth and dissolution of the dendritic conduction channel of the Yanson point contact [14]. Similar behaviour is also typical for copper point contacts obtained by mechanical manipulations in the framework of the “break junction” technique [42] and for silver point contacts obtained by electrochemical deposition [43]. In the process of numerous cyclic changes, the quantum system of dendritic Yanson point contacts passes through its characteristic set of metastable quantum states. The appearance of one or another quantum state is characterized by a certain probability, which can be visualized as a conductance histogram of the dendritic Yanson point contacts (Figure 9) [1,14]. The conductance histogram, similarly to a human fingerprint, is an individual characteristic of the system of dendritic Yanson point contacts under certain experimental conditions. A change in the conditions leads to a change in the set of metastable energy states of the system, which can be detected and recorded in the process of the cyclic switchover effect. Processing these data and plotting a conductance histogram provides a practical implementation of the quantum mechanism of selective detection in gaseous and liquid media [1], which underlies the operation of a quantum point-contact sensor.

Comparing the capabilities of the above research methods, we can see that optical and physicochemical methods for studying point-contact sensor systems successfully complement each other. At the same time, Raman spectroscopy makes it possible to obtain detailed information about the presence and properties of materials with a semiconductor type of conduction, impedance spectroscopy effectively describes the behaviour of the surface of the metallic conductors and the metal–electrolyte interface, whereas electrophysical measurements allow studying electrical conductance of objects with direct electrical conduction in the bulk and record quantum states of dendritic Yanson point contacts. A combination of optical and physicochemical methods of research makes it possible to finely control the state of the surface of the sensing element of a quantum sensor, which is the conduction channel of the Yanson point contact. The presence of semiconductor layers or adsorbed atoms that overlap and block the active centers on the electrode surface can affect the cyclic switchover effect. Adsorption of atoms of the analyzed medium on the conduction channel of a dendritic Yanson point contact significantly affects the set of metastable quantum states of the system and results in a change in the profile of the chronoresistogram which is the primary output characteristic of the sensor device [1].

Studying the kinetics of these mechanisms will lay the groundwork for future development of quantum sensor technologies by illuminating the subtle mechanisms by which analytes affect conductance quantization. These processes can be unveiled by manipulating the composition of the material under study while employing optical and physicochemical experimentation techniques. The versatile device we designed makes sure that these requirements are fully met. All of this demonstrates how well the devised tool works for analyzing the needle–electrolyte–anvil system of nanostructured quantum sensors.

## 5. Conclusions

In this work, a portable device for multipurpose research on dendritic Yanson point contacts and quantum sensing was developed and manufactured. With the aid of this apparatus, one can produce dendritic Yanson point contacts and investigate their quantum characteristics, which are amply demonstrated in the electrochemical cyclic switchover effect. The device tests showed that it was possible to gather data on the composition and characteristics of the synthesized substances, on the electrochemical processes that influence the synthesis of dendritic Yanson point contacts, as well as on the electrophysical processes that provide information on the quantum nature of the electrical conductance of these point contacts.

Due to its small size, the device can be easily incorporated into a micro-Raman spectrometer setup, providing a variety of options to tackle novel research issues related to Yanson point contacts.

The testing of the device also showed that it was possible to observe the electrochemical cyclic switchover effect and use the quantum mechanism of selective detection in fluids that were both gaseous and liquid. It should be noted that the operation of the quantum point-contact sensor is essentially different from that of the conventional conductive sensors. As demonstrated in Ref. [1], observation of the cyclic switchover effect allows use of the quantum sensory tool to selectively detect any gas agents, including noble gases. Quantum point-contact sensors demonstrate absolute selectivity and a very high sensitivity (see Refs. [1,12]). The main analytical instrument for the analysis of the studied medium is the conductance histogram of dendritic copper Yanson point contacts. In our research, we have shown that the device we propose for the first time provides the possibility of both observing the cyclic switchover effect, as well as the conductance quantization, and getting the data needed for the conductance histogram determination. It means that the device can operate in the framework of the methodology based on the quantum mechanisms of detection in liquid and gaseous media described in detail in Ref. [1]. This allows the design of a system that can be used as a prototype for developing a quantum sensor that will serve as the foundation for cutting-edge sensor technologies based on quantum mechanisms and principles of detection in gaseous and liquid media. The proposed device is a relevant development to be applied to research into the atomic-scale junctions, single-atom transistors, and any relative subjects [44,45].

## Figures and Tables

**Figure 1 nanomaterials-13-00996-f001:**
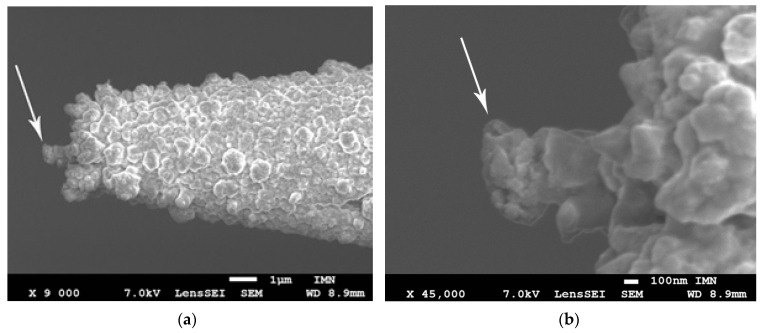
(**a**) Scanning electron micrograph (SEM) of a Cu needle tip after passing electric current through the needle–electrolyte–anvil system. Distinct areas of growing dendrites are seen. The arrow points to the most developed dendrite that will form a point contact. (**b**) SEM picture of the dendrite.

**Figure 2 nanomaterials-13-00996-f002:**
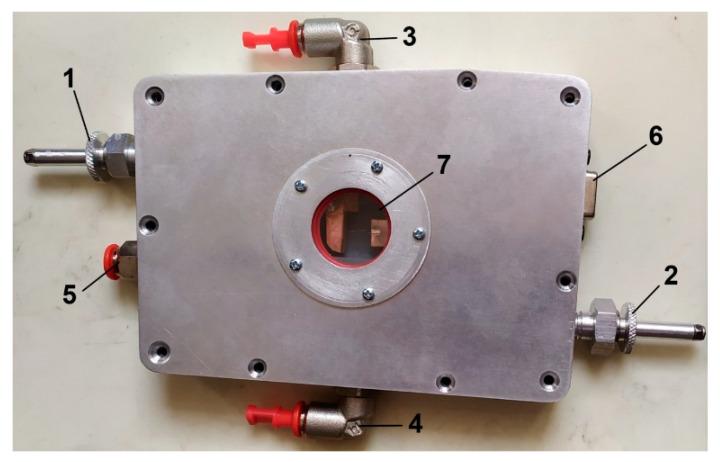
The prototype of the device for multipurpose research on dendritic Yanson point contacts and quantum sensing. 1, 2, electrode manipulators; 3, 4, gas connectors; 5, electrolyte capillary connector; 6, electrical connector; 7, optical window.

**Figure 3 nanomaterials-13-00996-f003:**
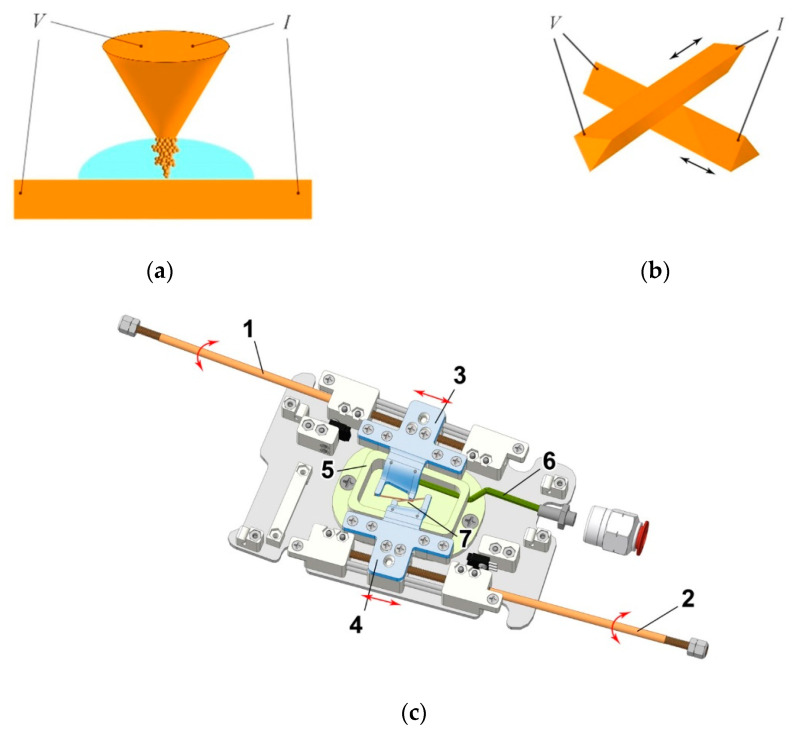
Schematic image of cartridge, as part of the laboratory prototype of the device for multipurpose research on dendritic Yanson point contacts and quantum sensing, which allows the creation of Yanson point contacts using both the needle–anvil and Chubov displacement techniques. (**a**) Schematic view of the dendrite growth in the framework of the needle–anvil method. (**b**) Schematic view of the Chubov displacement technique. (**c**) Cartridge. 1, 2, manipulator rods; 3, 4, electrode holders; 5, cuvette; 6, capillary; 7, production of a Yanson point contact using the Chubov displacement technique. *I*, current, *V*, voltage.

**Figure 4 nanomaterials-13-00996-f004:**
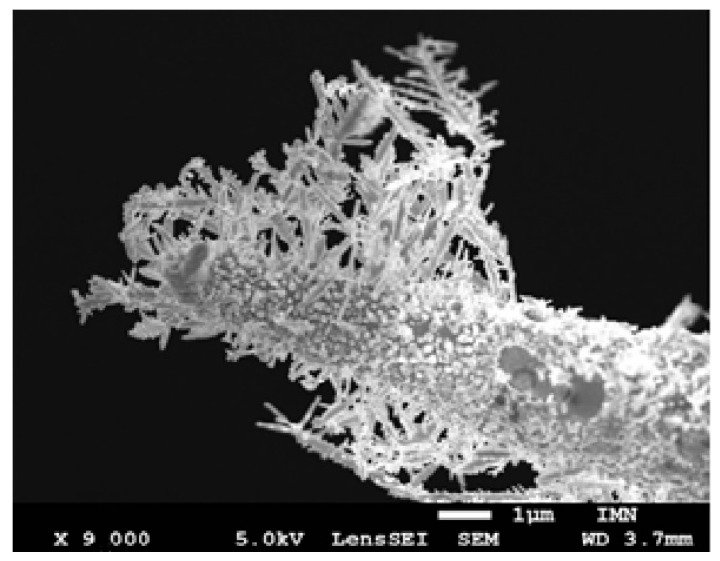
Needle electrode covered with a dendrite coat after experiment in atmospheric air. The needle electrode and the anvil electrodes were spaced within a few micrometers from one another at the start of the experiment.

**Figure 5 nanomaterials-13-00996-f005:**
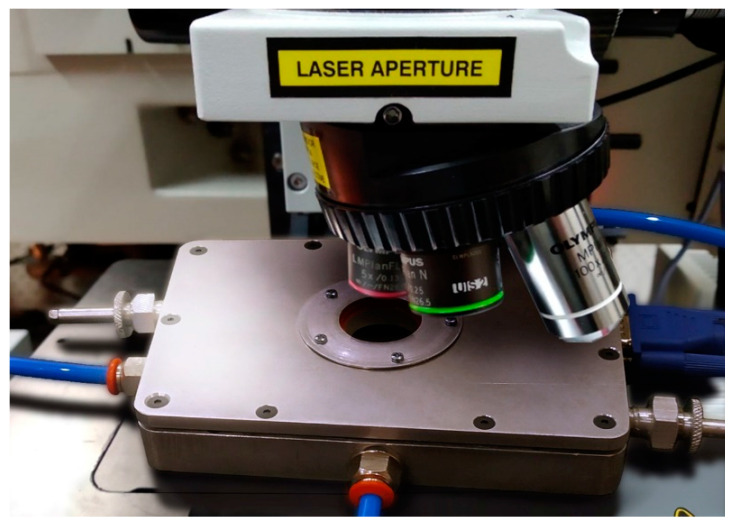
General view of the device for multipurpose research on dendritic Yanson point contacts and quantum sensing. The device is placed below the microscope of a Renishaw spectrometer.

**Figure 6 nanomaterials-13-00996-f006:**
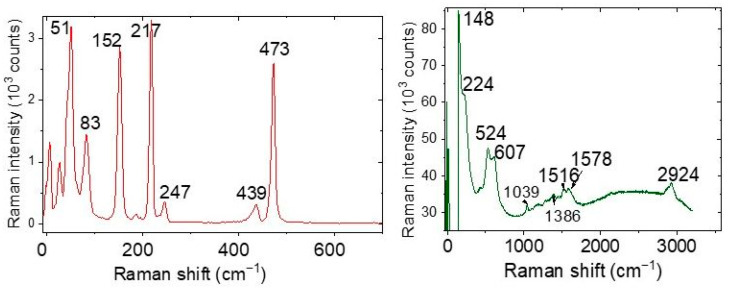
Typical Raman spectra taken on different tip spots using laser excitation at 633 nm.

**Figure 7 nanomaterials-13-00996-f007:**
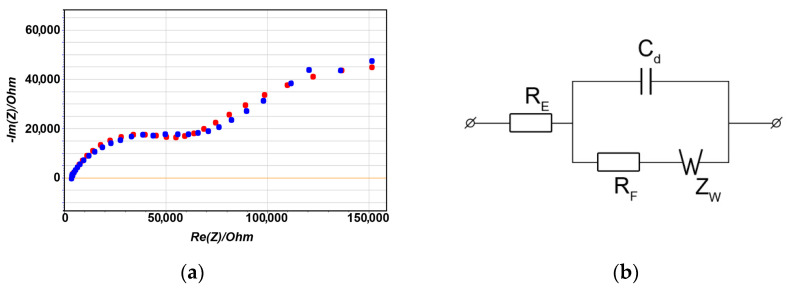
Impedance hodograph (**a**) and equivalent circuit (**b**) of the system “copper needle in a solution of copper sulphate with a concentration of 0.01 mol/dm^3^. The auxiliary electrode is a massive copper plate. The specific Faraday resistance is (*R*_F_)_sp_ = 0.75 Ohm cm^2^. The specific capacitance of the double layer is (*C*_d_)_sp_ = 2∙10^−5^ F/cm^2^. The electrolyte resistance is *R*_E_ = 4000 Ohm. The specific Warburg constant is *W*_F_ = 25 Ohm cm^2^ s^−0.5^.

**Figure 8 nanomaterials-13-00996-f008:**
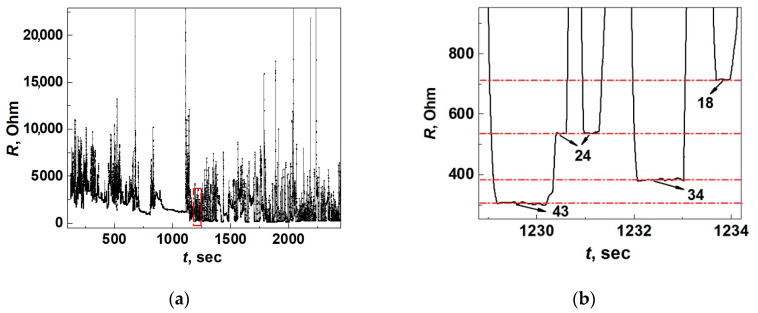
Dependence of resistance *R* on time *t* which reflects the transformations in the dendritic Yanson point contacts during the cyclic switchover effect in the ambient air. (**a**) General view of the curve *R*(*t*). (**b**) A large-scale image of the dependence *R*(*t*) in the area within the red rectangular section shown in (**a**). The number of conductance quanta that are associated with each specific metastable condition are indicated by arrows in (**b**).

**Figure 9 nanomaterials-13-00996-f009:**
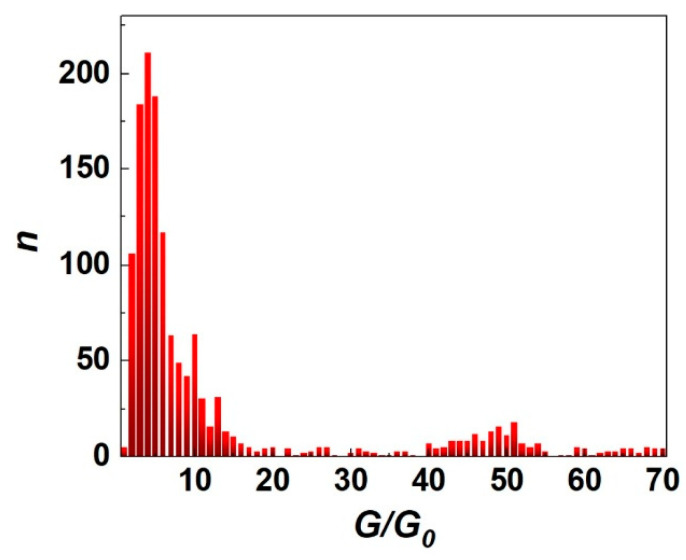
Conductance histogram of dendritic copper Yanson point contacts grown in the drop of ultrapure water in the ambient air. *G* is the conductance, *G*_0_ is the conductance quantum, *n* is the number of counts.

**Table 1 nanomaterials-13-00996-t001:** Calculated and experimental vibrational frequencies of CuO and Cu_2_O reported from literature (†, Refs. [27,28,31,32]) and found in this work (*). Units are given in cm^−1^. s: strong intensity; m: medium; w: weak.

Calculated Frequencies	Mode Symmetry	Experimental Frequencies (†)	Tip Frequencies (*)
CuO			
153.1	A_g_	147 (s)	148, 152 (s)
172.2	A_u_	163	
203.8	A_g_	220 (s) also in Cu_2_O	217, 224 (s)
293	A_g_	296 (s)	
323.2	A_g_	324	
346.5	A_u_	344 (w)	
449.	A_u_	438, 444	439(w)
520.9	A_g_	515	524, 532 (m)
599.8	A_g_	586	607 (m)
Cu_2_O			
85	E_u_	88 (w)	83 (m)
147.3	T_1u_	149 (s)	148, 152 (s)
341.	A_2u_ (silent)	350	
500.5	T_2g_	515 (w)	524 (m)
614.7	T_1u_	609, 628 (s) also in CuO	607, 629 (m)

## Data Availability

Not applicable.

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
