# Peer review of "Portable Device for Multipurpose Research on Dendritic Yanson Point Contacts and Quantum Sensing"

_nanomaterials, 2023, doi:10.3390/nano13060996_

Round 1

Reviewer 1 Report

The paper titled “Portable device for multi-purpose research on dendritic Yanson point contacts and quantum sensing” demonstrates the design and experimental results of a portable chemical sensor for gaseous and liquid media utilising the high sensitivity nature of Yanson point contacts. The manuscript is well structured and it seems like the sensor is almost ready for its potential application. Therefore, I recommend publishing this manuscript on MDPI Nanomaterials if the authors can address the following points:

1. The authors need to mention how they calibrate the distance forming the Yanson point contact and how this distance needs to be controlled. The authors mention a step size of 25nm for the manipulator rod, is this sufficient for the required resolution? Why not use a piezo stage?

2. The authors showed the results of the contacts growth in distilled water. But to show that the device can be used for sensing, at least one other analyte should be measured for comparison.

Author Response

Dear Referee,

we would like to thank you for your professional consideration of our submission and useful remarks concerning our manuscript. We have thoroughly addressed point-by-point all your comments with the associated changes listed in this letter. All changes in the manuscript are highlighted in yellow. We sincerely hope that we have successfully dealt with all the questions so that the manuscript can be accepted for Nanomaterials in its revised form.

Comments 1:  The authors need to mention how they calibrate the distance forming the Yanson point contact and how this distance needs to be controlled. The authors mention a step size of 25nm for the manipulator rod, is this sufficient for the required resolution? Why not use a piezo stage?

Answer:

As we described in the manuscript (see lines 291-294 in the revised manuscript): “The launch of the cyclic switchover effect can be started both at the moment of formation of direct electrical contact and after mechanically breaking the initially created contact and placing the needle at the minimum distance from the anvil.” We added an additional explanation after this sentence:

“Let us first consider the latter case. After mechanically breaking the initially created contact, one can move the needle electrode away from the anvil with a step of 25 nm. One step of the manipulator is equal to one turn of the rod at the gearbox and it provides a distance between the electrodes which is enough to launch the cyclic switchover effect. We found that the growth of a single dendrite can preferably take place at distances up to hundreds of nanometers between the needle and anvil electrodes (Figure 1). Within this range, it doesn’t matter what distance is chosen to launch the cyclic switchover effect. It should be noted, that the electrodes, especially the needle, are thoroughly treated and prepared for experiments, that is their side surfaces are clean and smooth and don’t contain impurities or inhomogeneities that can provoke dendrite growth.

In the former case, one can create a Yanson point contact using the methodology described elsewhere (see, for example, Ref. [12, 15, 16, 18, 19]). After the creation of the point contact, it is necessary to check its mechanical stability, and then, if it is ok, it is possible to launch the cyclic switchover effect by sweeping current through the contact. This technique is thoroughly described in Ref. [22].”

When designing the presented portable device, we considered different types of possible technology solutions, including a piezo driver, and took into account the portability, simplicity of the technique, as well as the economical efficiency of the device production. For example, the price of another original quantum point-contact sensor device, that we have already successfully developed and tested in real clinical conditions (see http://qs.net.ua/results_eng.html) is less than 100 Euro. Low prices along with the unique possibility of quantum point-contact sensors in medical diagnosis (see Ref. [2-3]) create very good competitive advantages for quantum sensor devices. Piezo driver demonstrated good qualities during the investigations of conductance quantization and quantum shell effect employing break junction technique. See, for example, Ref. [42]. But on the whole, the technology developed by Bobrov et al. (Ref. [18]) is simpler, cheaper, and offers more functionality. For example, it allows simultaneously using the “needle-anvil” method and the Chubov displacement technique (Ref. [17]) during one experimental cycle. This was successfully demonstrated during the investigations of anisotropy of electron-phonon interaction in organic conductors. See, for example, G. Kamarchuk et al. J. Phys.: Condens. Matter, 1994, 6, 3559-3566. In theory, it is possible to realize the Chubov displacement technique, which is the most effective method in the Yanson point-contact spectroscopy, using the piezo approach. But it requires two or more piezo elements, leads to complex solutions, and increases the cost of the device. Taking all this into account, we have chosen a mechanical approach presented in our manuscript.

Comments 2:  The authors showed the results of the contacts growth in distilled water. But to show that the device can be used for sensing, at least one other analyte should be measured for comparison.

Answer:

In our manuscript we considered the most difficult situation of launching the cyclic switchover effect using distilled (pure) water. The cyclic switchover effect is the basis for realization the quantum mechanism of selective detection in dynamic mode (Ref. [1]). In this case, water is needed to provide the possibility of detecting various agents at a minimum concentration level. The advantages of pure water are also creation of the necessary conditions to avoid potential influence of impurities and side effects on the results of quantum point-contact sensor detection. If one can launch the cyclic switchover effect in pure water, there will be no problem observing it in any electrolyte using the same technique. An electrolyte allows launching the cyclic switchover effect much easier than pure water does, see, for example, Ref. [14]. We have already performed several investigations with different electrolytes using the device presented in the manuscript and obtained results that are being used now for the preparation of other publications.

Sincerely yours,

The authors

Reviewer 2 Report

Paper refers to development of portable device for research on dendritic Yanson point contacts and quantum sensing. Paper fits the scope of the journal however it requires som improvements.

1. Conclusions are rathere short. Author did not give any numerical paremeters that were achieved to confirm novelty.

2. Figure 6 - there is lack of units.

3. Type of Renishaw spectrometer should be given as well as temperature of detector.

4. Table 1 - Authors should give not only positions of the line but also their intensity, at least strong, medium,weak.

Author Response

Dear Referee,

we would like to thank you for your professional consideration of our submission and useful remarks concerning our manuscript. We have thoroughly addressed point-by-point all your comments with the associated changes listed in this letter. All changes in the manuscript are highlighted in yellow. We sincerely hope that we have successfully dealt with all the questions so that the manuscript can be accepted for Nanomaterials in its revised form.

Comments 1:  Conclusions are rather short. Author did not give any numerical parameters that were achieved to confirm novelty.

Answer:

We have added more data in conclusions to underline the novelty of the proposed device (see lines 542-554 in the revised manuscript):

“It should be noted that the operation of the quantum point-contact sensor is essentially different from that of the conventional conductive sensors. As demonstrated in Ref. [1], observation of the cyclic switchover effect allows using the quantum sensory tool to selectively detect any gas agents, including noble gases. Quantum point-contact sensors demonstrate absolute selectivity and a very high sensitivity (see Ref. [1, 12]). The main analytical instrument for the analysis of the studied medium is the conductance histogram of dendritic copper Yanson point contacts. In our research, we have shown that the device we propose for the first time provides the possibility of both observing the cyclic switchover effect, as well as the conductance quantization, and getting the data needed for the conductance histogram determination. It means that the device can operate in the framework of the methodology based on the quantum mechanisms of detection in liquid and gaseous media described in detail in Ref. [1].”

Comment 2:  Figure 6 - there is lack of units.

Answer:

We have updated Figure 6. 

Comment 3:  Type of Renishaw spectrometer should be given as well as temperature of detector.

Answer:

We inserted information about the type of Renishaw spectrometer and temperature of the detector (see lines 350-356 in the revised manuscript):

“Micro-Raman experiments were performed in the device with a Renishaw inVia Raman Microscope spectrometer (Figure 5) including a detector working with a thermoelectric chip cooled at -70°C. Measurements were made at room temperature in the wavenumber range between 500 and 4000 cm−1. Laser excitation λexc = 633 nm (1.96 eV) was provided by a helium-neon laser. The detector allowed registration of the low-frequency part of the spectra (10-150 cm−1) and some anti-Stokes Raman lines by using a Bragg reflector for this wavelength.”

Comment 4:  Table 1 - Authors should give not only positions of the line but also their intensity, at least strong, medium, weak.

Answer: We have updated Table 1.

Sincerely yours,

The authors

Reviewer 3 Report

In this manuscript, the author reports, ‘Portable device for multi-purpose research on dendritic Yanson point contacts and quantum sensing”. The authors should address the following questions before getting a possible publication.

 Recommendation: Minor revisions are needed as noted.

1.     The novelty of the present article should be discussed in the last paragraph of the Introduction section.

2.     Abbreviations should be defined at their first instance.

3.     The author should write the purpose for each test in one/two sentences (in brief) before explaining the results of the characterization techniques.

4.     The formatting and grammatical errors in the article need to be checked carefully.

5.     The Raman spectroscopy analysis should be cited with relevant references.

6.     The authors have cited relevant references in the Introduction section; however the manuscript needs to be highlighted with recent reports further to broaden the impact.

Author Response

Dear Referee,

we would like to thank you for your professional consideration of our submission and useful remarks concerning our manuscript. We have thoroughly addressed point-by-point all your comments with the associated changes listed in this letter. All changes in the manuscript are highlighted in yellow. We sincerely hope that we have successfully dealt with all the questions so that the manuscript can be accepted for Nanomaterials in its revised form.

Comment 1:  The novelty of the present article should be discussed in the last paragraph of the Introduction section.

Answer:

We have added a short explanation of the novelty of our article (see lines 114-125 in the revised manuscript):

“To our knowledge, the proposed device is a prototype developed for the first time for realization of a quantum sensor whose operation is related to the new mechanism based on conductance quantization. The prospective pilot sample can be used to realize a series of new original quantum and sensing properties of Yanson point contacts and new effects arising from them. The quantum nature of the point-contact sensor response based on conductance quantization realized in the proposed device will ensure a successful search for novel quantum mechanisms of sensor analysis which have no analogues in the field of sensor technique, as well as their subsequent study, development, and implementation. The simplicity of construction and low cost of the point-contact sensor device will also stimulate its wide application in sensor research into the detection of different gaseous agents. It will establish a solid background for the development of principally new nanomaterial‑based technologies and devices.”

Comment 2: Abbreviations should be defined at their first instance.

Answer: We have carefully checked the text of the manuscript. We deleted the abbreviation (IMN) in the acknowledgement section.

Comment 3: The author should write the purpose for each test in one/two sentences (in brief) before explaining the results of the characterization techniques.

Answer:

We have added some clarifying comments to the text to explain the purpose of each test:

Lines 317-320: “The aim of the first test concerning device functionality is visualization of the processes under investigation and demonstration of the possibility to obtain direct information about the synthesis products and transformation of the materials in the working area.”

Lines 389-391: “The next portion of data that can be obtained using the proposed device is information about the nature of the homogeneous and heterogeneous physical-chemical processes taking place during the cyclic switchover effect.”

Lines 451-453: “The purpose of the third test is to demonstrate the device operation in the regime of cyclic switchover effect and obtain data about conductance quantization that can be used for detection in liquid and gaseous media.”

Comment 4: The formatting and grammatical errors in the article need to be checked carefully.

Answer: We have carefully checked the text of the manuscript and corrected the errors.

Comment 5: The Raman spectroscopy analysis should be cited with relevant references.

Answer: We have added new relevant references to the text. See Refs. [27, 28, 31, 44].

Comments 6: The authors have cited relevant references in the Introduction section; however, the manuscript needs to be highlighted with recent reports further to broaden the impact.

Answer: We have added new relevant references to the introduction. We also made a concluding remark about possible additional applications of the proposed device and reflected it in the abstract. The new references added to the revised manuscript are Refs. [9-11, 22, 31, 44-45].

Lines 557-558: The proposed device is a relevant development to be applied to research into the atomic-scale junctions, single-atom transistors, and any relative subjects [44, 45].

Sincerely yours,

The authors

Round 2

Reviewer 1 Report

The revised version has indeed improved the clarity and quality of the manuscript.